# Assessing the effectiveness of COVID-19 vaccine lotteries: A cross-state synthetic control methods approach

**Sam Fuller**[1][☉]*, **Sara Kazemian**[1][☉], **Carlos Algara**[2][☉], **Daniel J. Simmons**[3][☉]

**1** Department of Political Science, University of California, Davis, Davis, California, United States of America, **2** Division of Politics & Economics, Claremont Graduate University, Claremont, California, United States of America, **3** Department of Political Science & International Relations, Saint Michael's College, Colchester, Vermont, United States of America

☉ These authors contributed equally to this work.
* sjfuller@ucdavis.edu

## Abstract

Vaccines are the most effective means at combating sickness and death caused by COVID-19. Yet, there are significant populations within the United States who are vaccine-hesitant, some due to ideological or pseudo-scientific motivations, others due to significant perceived and real costs from vaccination. Given this vaccine hesitancy, twenty state governors from May 12th to July 21st 2021 implemented some form of vaccination lottery aiming to increase low vaccination rates. In the aftermath of these programs, however, the critical question of whether these lotteries had a direct effect on vaccination remains. Previous literature on financial incentives for public health behaviors is consistent: Financial incentives significantly increase *incentivized* behaviors. Yet, work done specifically on state vaccine lotteries is both limited in scope and mixed in its conclusions. To help fill this gap in the literature, we use synthetic control methods to analyze all 20 states and causally identify, for eighteen states, the effects of their lotteries on both first-dose and complete vaccination rates. Within those eighteen states, we find strong evidence that all but three states' lotteries had positive effects on first-dose vaccination. We find for complete vaccinations, however, over half the states analyzed had negative or null effects. We explore possibilities related to these mixed results including the states' overall partisanship, vaccine hesitancy, and the size of their lotteries finding null effects for each of these explanations. Therefore, we conclude that the design of these programs is likely to blame: Every state lottery *only incentivized first-doses* with no additional or contingent incentive based on a second dose. Our findings suggest that the design of financial incentives is critical to their success, or failure, but generally, these programs can induce an uptake in vaccination across diverse demographic, ideological, and geographic contexts in the United States.

**Data Availability Statement:** Data and replication code is held on Harvard Dataverse (https://doi.org/10.7910/DVN/K1XX02).

**Funding:** The authors received no specific funding for this work.

**Competing interests:** The authors have declared that no competing interests exist.

## Introduction

Health officials and politicians alike emphasize the importance of herd immunity to protect individuals from COVID-19. The least risky path towards achieving such an outcome, or at least shifting the SARS-COV-2 virus to an endemic state, is by vaccination of a critical mass of the population. Disappointingly, shortly after the mass availability of vaccines in the United States and Europe in April and May of 2021, vaccination rates remained strikingly low. In the United States in particular, vaccination rates ranged from 42.6% first-dose on April 19th, 2021 (when vaccination became widely available in all 50 states to anyone 18 and over) to only 48.5% first-dose on May 12th, 2021 [1]. Given this widespread lack of vaccination, US governors across the country began offering financial incentive programs, including well-publicized lotteries, to encourage vaccination. On May 12th, 2021, Ohio became the first state to offer such a program through the "Vax-a-Million" lottery. The program aimed to encourage Ohioans to get vaccinated by offering entry to the lottery in exchange for receiving their first dose of a COVID-19 vaccine (or the only dose in the case of the Johnson & Johnson vaccine). Nineteen states, reported in Fig 1, followed Ohio's lead from May 20th to July 21st, 2021, implementing similar vaccine lottery programs. Massachusetts, for example, offered fully vaccinated citizens a chance to win $1 million in cash prizes. In Michigan, partially vaccinated residents could register to win $50,000 and enter a separate lottery for a chance at winning up to $2 million. In support of these incentive-programs, survey experiments by the UCLA COVID-19 Health and Politics Project showed 34% of unvaccinated individuals reported being more likely to get vaccinated if given a cash prize [2].

This begs the important question: Did financial-based incentive programs, such as the twenty state-sponsored lotteries, actually increase an individual's willingness to get vaccinated against COVID-19? Existing research on COVID-19 vaccine lotteries in US states provides mixed conclusions, and is largely focused on the Ohio "Vax-a-Million" case. For instance, Walkey, Law, and Bosch [3] cast doubt on the efficacy of these programs in bolstering vaccination rates. Using data from the Centers for Disease Control and Prevention (CDC) COVID-19 Vaccine Tracker to conduct an interrupted time series analysis, they show vaccination rates

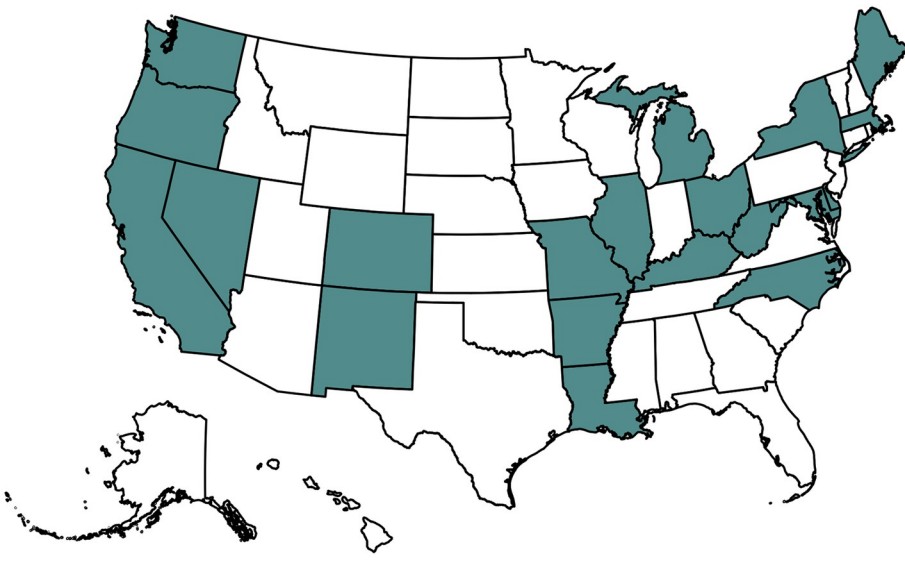

| COVID-19 Lotteries | Date Implemented |
|---|---|
| Ohio | 05/12/21 |
| Maryland | 05/20/21 |
| New York | 05/20/21 |
| Oregon | 05/21/21 |
| Arkansas | 05/25/21 |
| Colorado | 05/25/21 |
| Delaware | 05/25/21 |
| California | 05/27/21 |
| West Virginia | 05/28/21 |
| New Mexico | 06/01/21 |
| Washington | 06/03/21 |
| Kentucky | 06/04/21 |
| North Carolina | 06/10/21 |
| Massachusetts | 06/15/21 |
| Maine | 06/16/21 |
| Illinois | 06/17/21 |
| Louisiana | 06/17/21 |
| Nevada | 06/17/21 |
| Michigan | 06/30/21 |
| Missouri | 07/21/21 |

**Fig 1. Map and table of states that conducted COVID-19 lotteries.**

did not significantly increase after Ohio announced their lottery. In contrast, Brehm, Brehm, & Saavedra [4] argue "vaccine lotteries may play a role in inducing the marginal patient to get vaccinated, and may do so at a reasonable cost per marginal vaccination."

Given this mixed evidence and relatively narrow focus on the Ohio case, we contribute to this ongoing question and policy debate by leveraging the causal-identification approach of the synthetic control method. Specifically, we use this approach to examine the extent lottery programs increased vaccination rates across eighteen states (we were unable to generate accurate controls for two states). While we are not the only researchers to use a synthetic control model to assess this question [4–7], we are the first to apply this method so widely to each state that conducted a vaccine lottery. This focus on *all* vaccine lottery states helps us increase our external validity and our analysis of both first-dose and complete vaccination trends. Doing so enables us to shed an important light on the structure of these policies and their resulting outcomes. Specifically, we find lotteries had mostly positive effects (16 of 18 states) for increasing first-dose vaccination, but mostly negative or null effects (10 of 18 states) for complete vaccination. Given the structure of these programs, a single-dose enters an individual into the lottery, this finding is understandable, but highlights the need for policymakers to better design these lotteries in the future. Simply put, if incentives are only geared towards one specific behavior, expecting individuals to go beyond that specific behavior is unreasonable.

Furthermore, we also assess whether a state's partisanship, vaccine hesitancy, and the total prize-pool of the lottery have an effect on whether lotteries had positive, null, or negative effects. Importantly, because our analysis covers 18 states (six "red," three "purple," and nine "blue"), we have some leverage to explore these possibilities. We find that even though COVID-19 vaccination has polarized markedly along party lines, we find no relationship between a state's overall partisanship and the effects of a lottery on either first-dose or complete vaccination trends. Similarly, we find no relationship between vaccine hesitancy nor the size of the prize-pool and the effects a lottery had on vaccination rates. Overall, we contend that the single most critical determinant of the success of these lotteries is their incentive structure: financial incentives only motivate the behavior that is rewarded.

## Theory

### Financial incentive programs for public health

Financial incentive programs are an increasingly popular public health tool. These programs seek to encourage healthy behavior by rewarding individuals with some form of a monetary incentive if they participate in an incentivized behavior or task [8]. In the context of COVID-19, every state that conducted a vaccine incentive program offered entry into their lottery only if an individual received their first COVID-19 vaccine dose. In terms of incentives, the goal of these programs is to reduce the perceived costs of receiving a vaccine by providing individuals with the potential of a monetary award. In economic terms, the entry into a lottery could be perceived as a definite value, specifically an "expected benefit," calculated as the probability (ies) of winning × the value of prize(s). This value provided from the lottery may help reduce some of these costs if vaccine decisions are in fact based on immediate costs. Among the most commonly reported reasons for not getting a COVID-19 vaccine during the initial roll-out were concerns about potential loss of wages, an inability to take time off work, potential out of pocket costs, missing work because of side effects, and vaccine safety [9]. If a lottery program provides a large enough expected benefit to offset the realized or potential costs of getting the vaccine for any given individual, we should expect them to get their first-dose.

According to behavioral economics, lotteries should generally be effective in inducing behavioral changes because individuals place greater weight on small probabilities (like

winning a lottery). This phenomenon is why individuals may buy lottery tickets or gamble in general despite the small probability they will win a prize. Indeed, these programs are effective because they "encourage people who both under-estimate communicable disease risks and over-estimate their likelihood of winning a lottery prize" [5]. Likewise, lottery programs can be particularly attractive for individuals who are willing to accept financial risk. If financial risk is related to other types of risky behavior like declining a vaccine to prevent a communicable disease, then these programs may be well suited for those at "higher risk of engaging in the undesirable behavior" [10]. Consequently, policymakers and health officials often implement lottery-based incentive programs for healthcare interventions [11].

Previous literature on rewards-based programs and improved health outcomes, however, is mixed. For example while some show financial incentives increase willingness to engage in a given activity [12, 13], others show lottery-type incentives are less effective than other alternative types of interventions [14]. Existing evidence from COVID-19 lottery programs also provide mixed conclusions. Specifically, three studies find Ohio's vaccine lottery program was effective in increasing state-wide vaccination rates [5, 7, 15]. In contrast, an equal number of other studies cast significant doubt on the effectiveness of lottery-based programs [3, 16, 17]. Going beyond the Ohio case, this paper provides a comprehensive analysis of every existing COVID-19 lottery program and explores the extent to which these programs can increase an individual's willingness to get a vaccine.

## Partisan theories of COVID-19 attitudes

As highlighted earlier, the principal task of state-based vaccine lottery programs was to increase vaccination rates among vaccine hesitant populations. While previous work suggests financial incentive programs could work effectively in raising state vaccination rates, contemporary work on the broader COVID-19 pandemic suggests partisanship plays a large role in shaping COVID-19 attitudes. We highlight countering partisan theories arguing partisan predispositions should blunt the effectiveness of financial lottery incentives to raise vaccination rates in hopes of combating the spread of the ongoing COVID-19 pandemic.

Originating from the early phase of the COVID-19 pandemic in the United States, scholars note partisanship plays a key role in shaping pandemic mitigation efforts such as mask wearing, social distancing, and general support for public policies aimed at containing the spread of COVID-19 [18–25]. These scholars find that the American mass public are polarized in both their attitudes regarding the pandemic, and in the general support for efforts to mitigate the spread of the pandemic, with Republican partisans being less likely to endorse such efforts than their Democratic counterparts. Perhaps most salient for our purposes, and mirroring similar partisan cleavages found in other democracies [26–28], scholars consistently note Republican partisans are far less likely to state an intention of vaccination than Democratic partisans [29]. This partisan divide is also reflected in aggregate data on administered COVID-19 vaccine doses at the county-level, with more Republican counties lagging behind more Democratic counties in population vaccination rates [30]. These aggregate findings along with individual-level intentions from survey data are striking, given that Republican counties were disproportionately more likely to suffer from COVID-19 deaths than more Democratic counties [23, 31].

This issue polarization also spreads to general perceptions regarding the severity of the public health crisis caused by COVID-19. Indeed, recent work finds Democratic partisans are more likely than Republican partisans to view the pandemic as a threat to their personal well-being as well as the well-being of their immediate family members [22, 24, 25]. In a similar vein, work by Clements [32] notes Democratic and Republican partisans are polarized on their

general knowledge of the acute health effects of contracting COVID-19, with Democratic partisans being more knowledgeable about the health consequences of contracting COVID-19 than Republican partisans. Even in the purview of mobility and compliance with shelter-in-place orders, congruent studies find counties that voted more for President Trump in the 2016 presidential election exhibited less compliance with physical distancing [33, 34].

While the observation that there is a polarized, partisan public is not a novel one, scholars have argued polarization on public health attitudes relative to COVID-19 are a distinct issue-area given the lack of preexisting public health issue cleavages in the American mass public [18]. Algara et al. [18] note, "in the context of the United States' polarized political system, however, COVID-19 presents an uncommon instance of a politically salient issue on which voters have few, if any, existing considerations to guide their support for policies or specific behaviors." The key ingredient to the mass public polarizing on COVID-19 policies are the cues being broadcast by partisan elites. From the onset of the pandemic, both the rhetoric and policy responses espoused by Democratic and Republican elites were distinct and polarized from one another. Green et al. [35], using Twitter outreach from elites, empirically supports a consensus on the polarization of COVID-19 threat messaging. Specifically, they find Republican members of Congress place far less emphasis on the threat of public health posed by COVID-19 than Democratic members of Congress. At the presidential-level, this divide was also apparent, with the Trump administration largely seen as disregarding the recommendations of public health experts to recommend and implement COVID-19 mitigation policies and largely downplaying the overall public health risks of the pandemic [23, 36–38]. In terms of partisan variation in policy responses to the COVID-19 pandemic, Adolph et al. [39] note Democratic governors implemented COVID-19 mitigation policies, such as mask mandates, much more often than Republican governors. Similarly, Shay [40] finds that Democratic governors were 50% more likely to implement stay-at-home orders than Republican governors, a finding confirmed by Baccini [41].

Taken together, theories assessing diverging attitudes and behavior relating to the COVID-19 pandemic emphasize the role of partisan attachments in shaping this variation, with partisan elite cues reinforcing these diverging attitudes. In terms of the effectiveness of state-based lottery programs, the partisan polarization among elected officials provides some insights of why one should be skeptical regarding the effect of these lotteries on COVID-19 vaccination rates. First, scholars note that even cues from co-partisans emphasizing the importance of vaccination or other public health mitigation efforts may not sway the mass public [23, 42]. Indeed, this lack of persuasion is particularly true among Republican partisans, with previous work showing endorsements by President Trump and non-partisan elites such as the World Health Organization to be insufficient in causing changes to Republican vaccination intention [42]. Secondly, partisan attitudes about COVID-19 are largely consistent throughout the pandemic, suggesting attitudes and preferences regarding vaccinations may be sticky and impervious to the elite-driven lottery-based incentive structures employed by states [43, 44]. State lottery outreach efforts designed to boost vaccination rates among the mass public may face headwinds on the basis of partisanship, with Republicans in particular being resistant to such outreach efforts after a full year of having their pandemic outlook reinforced by co-partisan elites (i.e., Republican members of Congress, President Trump) [43].

## Evaluating competing theories beyond a single state

While our study is not the first to explicitly assess the effectiveness of state lottery COVID-19 vaccination programs, we note a few limitations in the standing literature. First, the few studies assessing the effectiveness of state lottery programs explain that these efforts were largely

unsuccessful or marginal in the face of a polarized and partisan public, while others found significant, positive effects of these programs [3, 5, 45, 46]. Overall, with this set of contradictory findings, there is a distinct lack of consensus regarding the effectiveness of these lottery programs. Second, these studies focus on a single case study by evaluating the effectiveness of the first lottery-based program, Ohio's "Vax-a-Million" lottery program, rather than considering all 20 state COVID-19 vaccination lottery programs. As Fig 1 shows, after Ohio's implementation of its lottery program another 19 states followed suit across the nation, providing for rich variation in evaluating the effectiveness of state lottery programs beyond a single state case. Third, while a handful of studies published do assess the effect of other state-based lottery programs in addition to Ohio, these studies leverage correlational research designs ill-equipped at estimating the causal effect of lotteries on vaccination rates [15, 17, 47]. For example, the studies by Law et al. [17] and Robertson et al. [15] leverage econometric descriptive designs that lack the necessary causal identification needed to estimate the causal effect of state lottery programs on COVID-19 vaccination rates [48, 49].

To build on previous studies, we use synthetic control methods to evaluate competing expectations with regards to the effectiveness of state lottery programs in increasing COVID-19 vaccination rates. We describe and estimate a series of synthetic control models, allowing us to explicitly identify the causal effect of lottery programs on vaccination rates within a given state for *each* context in which a state lottery policy was implemented. Building on pioneering work by [48], our design allows us to contribute on the effectiveness of policies facilitating public health vaccination by evaluating observed changes in vaccination over time in a given "treated" state with a lottery program relative to a synthetic control representing an "untreated" counterfactual state. Our design allows us to more causally evaluate competing theoretical expectations regarding the effectiveness of state lottery programs beyond a single state case study and using more complete data than previous synthetic control method studies.

## Research design

### Data sources

To test our competing arguments, we require data measuring both a state's vaccination rates and when a state implemented a statewide lottery program, if at all. To collect our outcome variable assessing a given state's vaccination rate over time, we rely on state-level data from the *US Centers for Disease Control & Prevention* (CDC), which tracks the number of administered first and second doses, signifying a complete vaccination regiment, administered in a given state on a given day. To measure our key indicator of the presence of a state-lottery program, we manually collect data on all 50 states, including the District of Columbia, and present the geographic distribution of these "treated" states implementing a COVID-19 vaccination lottery program in Fig 1. We also present the date of implementation in Fig 1 and find 20 states were treated with a COVID-19 vaccination lottery and 31 states (including D.C.) were not.

Our synthetic control design requires state-level covariates (weights) to construct synthetic counterfactuals/controls of our treated state units (i.e., states with a lottery program). We collect US Census data from the 2015–2019 American Community Survey to capture state demographic variables such as income, race, education, and percent of the population who are elderly. In addition, we collect data on estimated state vaccine hesitancy rates from the US Census' Week 29 Household Pulse Survey fielded from April 26th to May 10th, 2021. Vaccine hesitancy rates for states were calculated from this raw data, with each of the five responses' percentage of total survey respondents being its own variable. Respondents who reported not being vaccinated were asked "Once a vaccine to prevent COVID-19 is available to you, would you. . .," with five possible answers provided: "Will definitely get a vaccine," "Will probably get

a vaccine," "Unsure about getting a vaccine," "Will probably not get a vaccine," and "Will definitely not get a vaccine." We use these answer choices as five separate categories to measure vaccine hesitancy. This data was collected to mitigate the effect of some states being more or less prone to receiving a COVID-19 vaccine than others from the onset of the vaccination roll-out. Importantly, this data was collected immediately before the beginning of the first vaccine lottery in Ohio on May 12th, 2021. Descriptive statistics of all data used in our analyses can be found in S3 Table.

Taken together, our data allows us to implement a synthetic control design for comparing lottery-treated state vaccination rates with the vaccination rates of synthetic counterfactuals/controls of these same treated units under a scenario where no lottery was implemented.

## The synthetic control method

Examining the effects of statewide COVID-19 vaccine lotteries presents an ideal use-case for the synthetic control method. Our use of the synthetic control method is motivated by four considerations: First, in the United States there are both a large amount of states that had lotteries, 20, along with those that did not, 31 including Washington, D.C. Second, we may have reason to believe lotteries had significantly heterogeneous effects across states which would be obscured by traditional methods like a panel, fixed effects regression. Third, given so many states had lotteries, using an individual, difference-in-differences approach would limit us to conclusions about only one state, as previously mentioned. Finally, using matching on units with so many unobserved characteristics as US states is rife with error, misspecification, and a lack of causal identification.

In detail, this methodology leverages data from "control" states without a lottery in a fundamentally different way than standard methods. The synthetic control method does not equally weight every control unit, instead opting to generate a synthetic counterfactual/control state that is comprised of different, actually observed weighted control states. Importantly, this generation of control units is transparent, above and beyond standard methods: We use a set of well-known optimization functions (see S1 Appendix) to remove researcher input that could lead to selection bias.

The process of conducting the synthetic control method is as follows: We select a population (US states) to analyze and divide the population into treatment (states with lotteries, denoted by set *I*) and control groups (states without lotteries, denoted by set *J*). We then select both the time period for analysis (including dates before and after treatment occurred for each analyzed treated unit/state, denoted by set *T*) and covariates measured at the unit level (these can be static or variable over the time period, denoted by set *C*) that may influence our selected outcome of interest (COVID-19 vaccination rates). These covariates in our case are:

- Income per capita

- Percentage of the state's population 65+

- Percentage of the state's population with at least a bachelor's degree

- Percentage white

- Percentage Black

- Percentage Hispanic

- Population per square mile

- Vaccine hesitancy (five categories/covariates)

To summarize, we have $|I| = 20$ treated cases, $i = 1, \ldots, 20$, $|J| = 31$ control cases, $j = 1, \ldots,$ 31, and $|C| = 12$ state-level covariates, $c = 1, \ldots, 12$, that are measured over $|T| = 107$ days, $t = 1, \ldots, 107$ (April 19th to August 4th, 2021), with a pre-treatment period for each treated state denoted $1 \leq t < T_i$, where $T_i$ denotes the treatment date of unit $i$, and a post-treatment period $T_i < t \leq 107$.

We use this method by individually generating synthetic controls for all 20 of our treated units. In other words, we attempt to generate a synthetic state that was the same on both observed and unobserved characteristics, but did not conduct a COVID-19 vaccine lottery. We use a strictly predictive model to maximize the fit of the outcome variable (vaccination rates) between the observed treated state and the synthetic control state *before the lottery occurs*, $1 \leq t < T_i$. These synthetic controls are constructed by first determining which covariates, $C$, are most predictive of vaccination trends individually in each of our treated states, $i \in I$. Once these covariates have been weighted based on their importance in predicting vaccination rates in state $i$ we are then able to determine which control states, $j \in J$, are most similar to treated state $i$. Specifically, these control states are weighted by a $|J| = 31$-length vector of weights, $w_i$, generated in the previous step. Finally, the synthetic control is generated for state $i$, using $w_i$, resulting in a counterfactual state that is a weighted-average of all control states. As mentioned above, this synthetic control state attempts to mirror the pre-treatment trend of the observed, treated unit. Importantly, the synthetic control also replicates the values of important covariates and unobserved, confounding variables that exist (and are time-variant) before the treatment.

Calculating a treatment effect is comparatively simple. Similar to difference-in-differences designs, any disconnect between the two trend lines after the treatment can be considered a treatment effect and calculated using standard average differences (time-averaged daily-differences [TADD], in our case). Importantly, an additional benefit of this method, above and beyond difference-in-differences, is its ability to account for not only *time-invariant* unobservable confounders but also *time-variant* unobservable confounders. Given this model's construction, however, causal identification is not achieved in the same way as difference-in-differences (first-differences over time). Consequently, we follow the relatively simple approach of Abadie et al. [48] for causal identification. They show this identification requires 1) our observed states correspond closely to their synthetic controls before treatment, and 2) the "true" model for estimating the outcome of interest is similar and comparable across every state. The first of these requirements is satisfied empirically for eighteen of our twenty analyzed states, for both outcomes of interest, and the second is satisfied by restricting our analysis temporally and geographically: We only analyze US states and consider only the time shortly before and after the announcement of lotteries [48–50].

These analyses were conducted using the statistical software R version 4.1.3 and relied on the synthetic control methods package `tidysynth` version 0.1.0. Replication materials, including code and data, to replicate all analyses, tables, and figures are available in our repository on Harvard Dataverse, linked in our data availability section.

## Results

First, to illustrate how synthetic control methods operate, and how effectively they work on our sample of US States, we look to Fig 2. This figure contains the plots of both the observed (gray) and synthetic (purple) trend lines of six states with the first row (A-C) showing states' first-dose trends and the second row (D-F) showing states' complete-vaccination trends, with dashed lines in each denoting the date the state began their lottery (please see S2 Appendix for trend plots and control-weight plots for all states). Again, the observed trend line comes

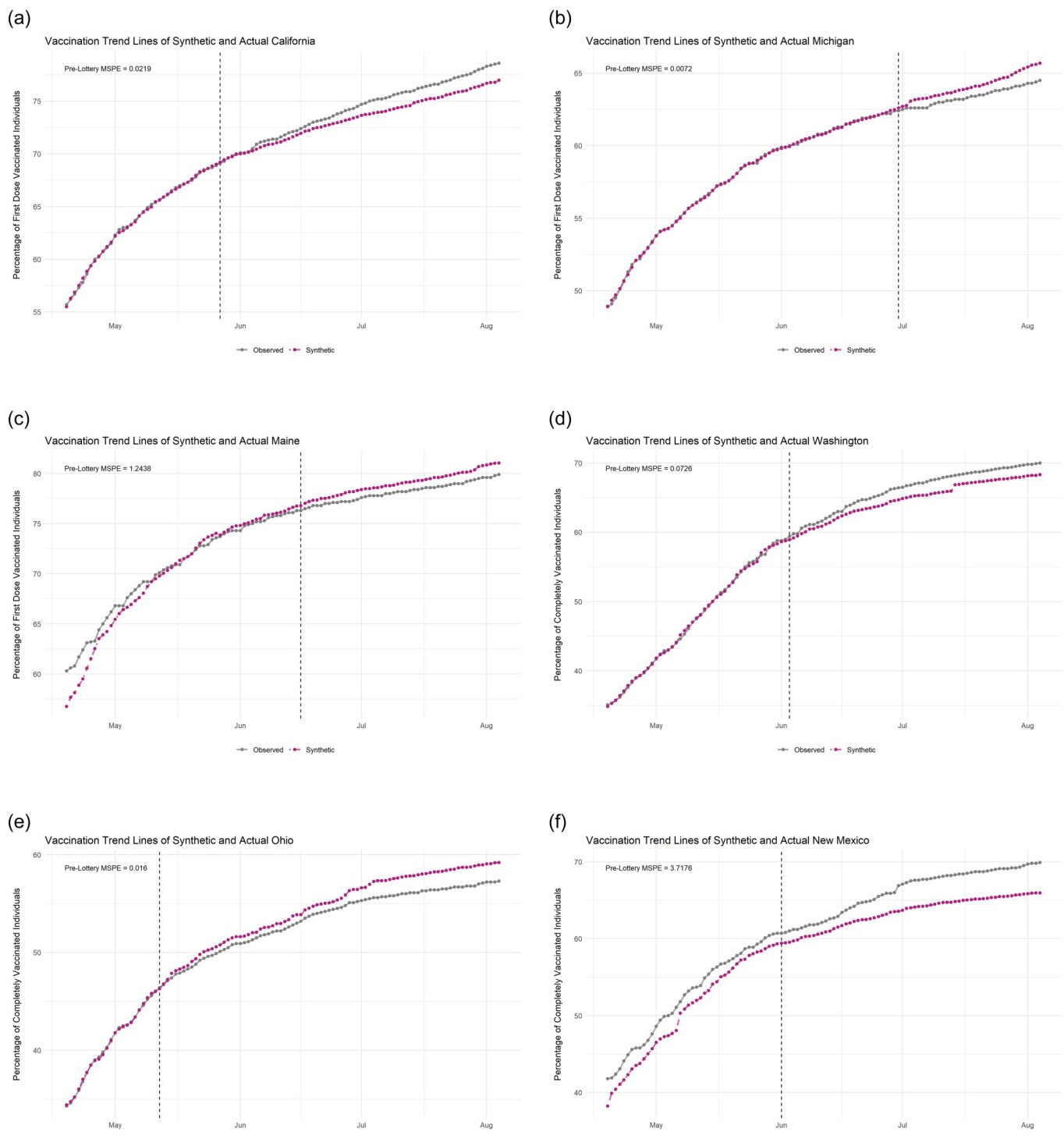

**Fig 2. Selected observed and synthetic vaccination rate trend lines by state.** A: California (May 27th) B: Michigan (June 30th) C: Maine (June 16th) D: Washington (June 3rd) E: Ohio (May 12th) F: New Mexico (June 1st).

directly from these states which conducted vaccine lotteries and the synthetic line is generated for each state separately as a counterfactual, i.e., what would the vaccination trend be if this given state *did not* conduct a lottery. The success of these methods to generate a believable counterfactual can be measured directly by the pre-treatment mean squared prediction error (MSPE), which are included in each plot. For tabular results of the fits of our models see S1 and S2 Tables, for first-dose and complete vaccination, respectively. In a more intuitive way, we can also examine visually how closely the trend lines match one another before the lottery (the dashed-lines). These trend lines match closely before treatment in panels A, B, D, and E. Given the great diversity of potentially unobserved, confounding characteristics of states that may effect vaccination trends, however, the methodology is ineffective in generating a believable counterfactual for two of the states in our sample: Maine (C) and New Mexico (F).

Similarly, we calculate post-treatment time-averaged daily differences (TADD) to determine whether a given state's lottery affected its vaccination rates, reported in Fig 3. We also analyze this effect visually by comparing the post-treatment (dashed-line) trend lines. If the observed (lottery) line is consistently above the synthetic (no-lottery) line, then the lottery had a positive effect in that state. If the opposite is true, the synthetic line is below the observed line, then we can say that state's lottery had a negative effect on vaccination rates. In Fig 2 we see California (A) and Washington's (D) lotteries had a positive effect on first-dose and complete vaccination, respectively. In contrast, Michigan (B) and Ohio (E) had negative lottery effects. Overall, trend plots help illustrate both the accuracy of synthetic controls pre-treatment and the effects of treatment, both positive and negative.

To present a fuller picture of the results of our analyses, we present the TADD effects of lotteries for every state, with the exception of our inaccurate synthetic controls (Maine and New Mexico), in Fig 3. Fig 3 shows the time-averaged daily differences, with 95% confidence intervals. Simply, positive lottery effects are points to the right of the dashed-line whereas negative effects are to the left, with null effects overlapping the origin. Overall, consistent with our expectation, lotteries had a positive effect on first-dose vaccinations for all but two states (Michigan and Arkansas), ranging from an increase of 0.11% in Massachusetts to 1.98% in Washington with Michigan being the only state with a negative effect at −0.71%. However, the complete vaccinations plot paints a very different picture, with nine states with negative effects, ranging from −0.12% in Illinois to −1.16% in Ohio, and eight states with positive effects, ranging from 0.25% in California to 2.15% in Oregon. Given this stark difference in the distribution of effects between first-dose and complete vaccinations, we explore possible alternative explanations in the following section.

## Possible explanations of the divergent effects of lotteries

**Partisanship.** To test our expectation that Republican states' lotteries would have reduced or even negative effects on vaccination rates, we explore whether there is any correlation between a state's 2020 Presidential Democratic vote-share and its TADD. In Fig 3, states are sorted and colored by President Biden's vote-share in the 2020 general election, with solidly Democratic states colored in blue, swing-states colored in purple, and solidly Republican states colored in red. For both first-dose and complete vaccination trends there appears to be no relationship between partisanship and a lotteries' TADD effect on vaccination. We also run two bivariate regressions (column 1) reported in Tables 1 and 2.

For first-dose TADDs, partisanship clearly has no explanatory relationship, with a small coefficient and a larger standard error along with a negative adjusted $R^2$. However, for complete vaccinations there is a stronger correlation, with the coefficient nearly twice the size of its standard error, and an adjusted $R^2$ of 0.149. However, it should be noted that this relationship

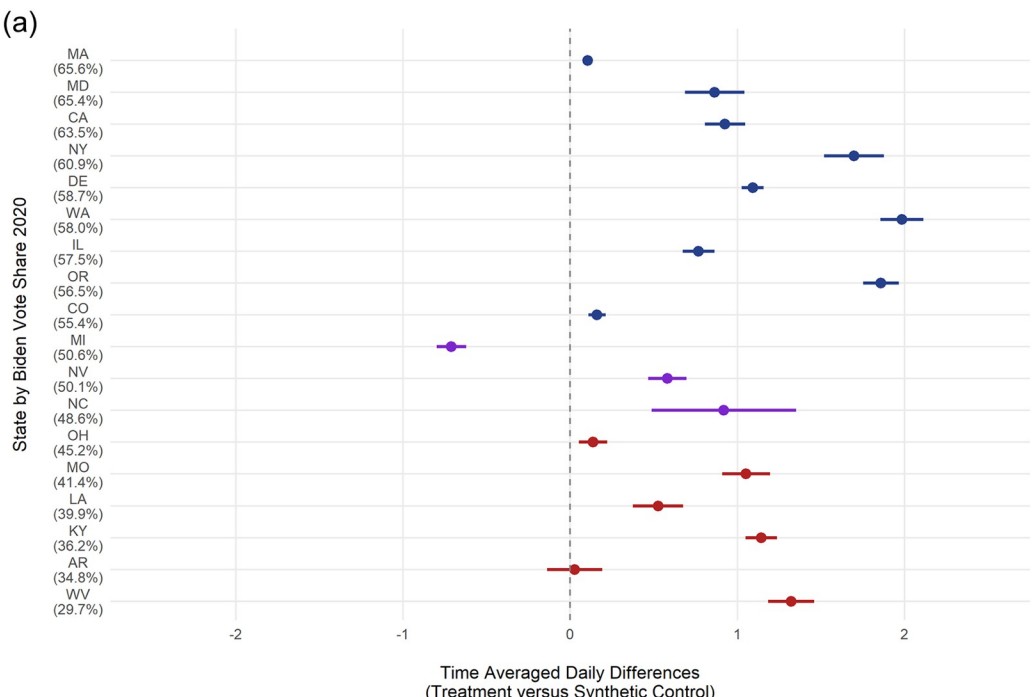

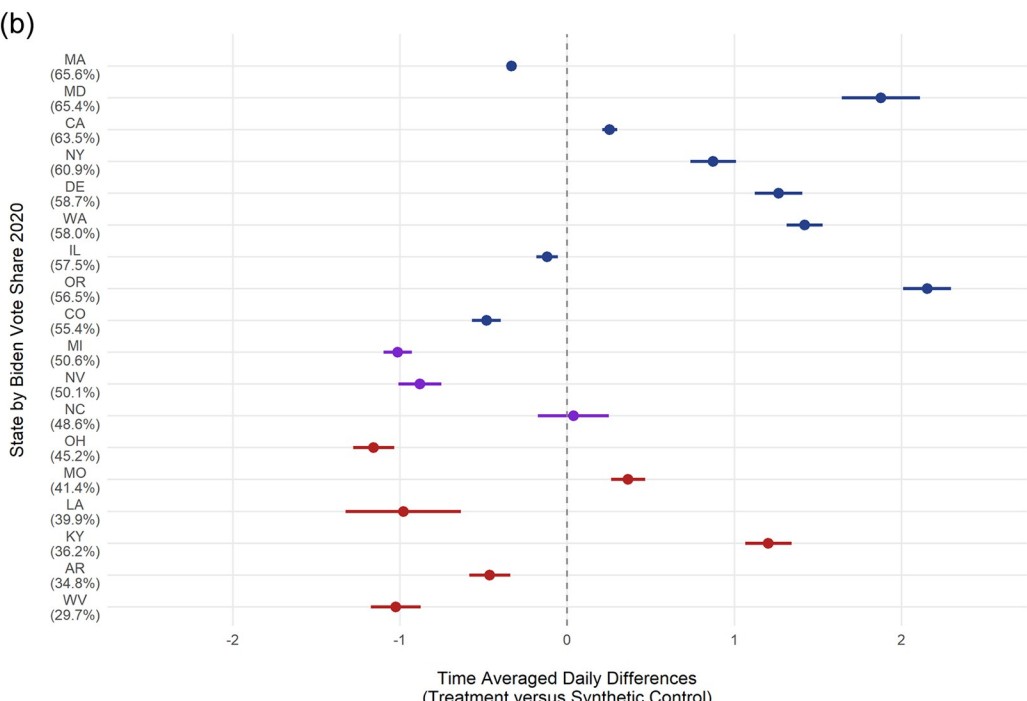

**Fig 3. TADD estimated effects of lotteries on vaccination rates by state and 2020 Biden vote-share.** A: First-Dose B: Complete.

**Table 1. TADD explanatory regression results (first-dose).**

|  | (1) | (2) | (3) | (4) | (5) |
|---|---|---|---|---|---|
| Democratic Vote-Share | 0.008 |  |  |  | 0.041 |
|  | (0.016) |  |  |  | (0.025) |
| Hesitancy |  | 2.898 |  |  | -1.443 |
|  |  | (5.074) |  |  | (10.188) |
| Lottery Value (in 100k $) |  |  | 0.001 |  |  |
|  |  |  | (0.006) |  |  |
| Lottery Value (Per Capita) |  |  |  | 0.092 | 0.015 |
|  |  |  |  | (0.103) | (0.205) |
| Constant | 0.411 | 0.107 | 0.734 | 0.615 | -1.642 |
|  | (0.817) | (1.230) | (0.186) | (0.225) | (2.709) |
| Observations | 18 | 18 | 17 | 17 | 17 |
| $R^2$ | 0.015 | 0.020 | 0.003 | 0.050 | 0.066 |
| Adjusted $R^2$ | -0.047 | -0.041 | -0.063 | -0.013 | -0.149 |

*Note*: NY does not have an estimated total lottery value and is thus dropped from regressions 3–5.

is outside conventional bounds of statistical significance and disappears when included in a saturated regression including other potential explanatory variables (detailed below, reported in column 5). Simply put, it appears that partisanship has no relationship to the outcomes of lotteries in a given state as measured by our synthetic control models. However, we stress that our results here only confirm that partisanship at the state level does not seem to be related to a lotteries' effect on vaccination. Partisanship is still quite possibly a distinct determinant of vaccination and vaccination intention at the individual level.

Our unexpected results relating to partisanship could be due to a few factors, particularly related to who the lotteries may be appealing to. First, we should expect that, according to a US Department of Health and Human Services report [9], that a significant portion of individuals were not vaccinated at the time due not to partisan polarization, but instead to the inability to

**Table 2. TADD explanatory regression results (complete).**

|  | (1) | (2) | (3) | (4) | (5) |
|---|---|---|---|---|---|
| Democratic Vote-Share | 0.043 |  |  |  | 0.005 |
|  | (0.022) |  |  |  | (0.017) |
| Hesitancy |  | -1.266 |  |  | -2.379 |
|  |  | (7.861) |  |  | (6.813) |
| Lottery Value (in 100k $) |  |  | -0.001 |  |  |
|  |  |  | (0.010) |  |  |
| Lottery Value (Per Capita) |  |  |  | -0.037 | 0.124 |
|  |  |  |  | (0.168) | (0.137) |
| Constant | -2.040 | 0.470 | 0.130 | 0.179 | 0.901 |
|  | (1.131) | (1.905) | (0.297) | (0.368) | (1.811) |
| Observations | 18 | 18 | 17 | 17 | 17 |
| $R^2$ | 0.199 | 0.002 | 0.000 | 0.003 | 0.182 |
| Adjusted $R^2$ | 0.149 | -0.061 | -0.067 | -0.063 | -0.006 |

*Note*: NY does not have an estimated total lottery value and is thus dropped from regressions 3–5.

get time off of work or the fear of losing their job due to vaccine side-effects. These worries can be considered, as the opposite to an expected benefit, as expected costs from receiving a vaccination. Given that lotteries can be perceived of as an expected benefit, then these lotteries are likely, if they are to motivate anyone to receive a vaccination, to affect these individuals. Thus, if these programs are targeting these non-partisan motivated individuals, then in general the partisanship of a state should have little to no impact on the direction of results from these lotteries.

**Lottery winnings, vaccine hesitancy, and incentive structure.** The share of the population that is on the margin of choosing to receive a vaccine may be an important explanatory variable for the effectiveness of a lottery. Particularly, as explained above, it is possible that these programs would directly target individuals who were essentially on the fence about getting a vaccine, as the expected benefit of the lottery may outweigh their perceived costs of receiving the vaccine. To test this possible explanation, we run a bivariate regression, reported in column 2, using the percentage of those "unsure" about receiving the vaccine in a given state to predict the lottery's effect for both first-dose and complete vaccinations. Again, we find no relationship between hesitancy and the effects of lotteries for either first-dose or complete vaccination.

Another strong contender for a factor that influences the effectiveness of lotteries is simply its total prize-pool per capita. In the context of an expected benefit, a larger prize-pool per capita should lead to a larger benefit and thus induce more individuals to get vaccinated. While in traditional rational-choice theory, individuals respond to this per capita expected benefit, we also consider the possibility that individuals instead respond to the absolute value of the lottery's prize-pool. Using approximate total lottery values from Barber and West [5], we run bivariate regressions, reported in columns 3 and 4, using both the total value (in 100k US dollars) and the per capita value of lotteries to predict the lotteries' first-dose and complete effects. Somewhat surprisingly, we find no relationship between the size of the lottery, in both absolute and per capita terms, and the estimated effect of said lottery.

Finally, we assess all of these potential explanations in a saturated regression, including all factors (only per-capita lottery value is used). Again, as above, we find no evidence of any of these variables being linked to our recovered causal effects of lotteries for either first-dose or complete vaccinations. Overall, it seems that all of our potential, measured explanations are unrelated to the results that we recovered.

While our results are contrary to our expectations regarding the factors that we assessed, it is consistent with both previous research on financial incentives and the design of these lotteries. First, our only noticeable explanation for the divergent effects of lotteries is whether we are assessing first-dose or complete vaccination rates, where there is a dramatic shift in the amount of positive versus negative effects between the two. Simply put, this difference in effects between first-dose and complete vaccinations is likely, at least in part, due to the incentive structure of *every one* of these lotteries: Individuals were entered after their first-dose, with no incentive for them to complete their series. Expecting individuals to participate in behavior above and beyond that which is directly incentivized is thus, misguided. It would seem that the major determinant for the effectiveness of these lotteries is simply their structure: Individuals will only respond to behaviors that are directly incentivized. Despite these initial findings, future work should interrogate, beyond state-level partisanship, vaccine hesitancy, and program structure, why the effects of lotteries had such divergent effects in different states. In particular, research should examine individual-level determinants of vaccination, with a focus on the interaction between partisanship and financial incentives.

**Limitations in analyzing potential explanations.** Importantly, it should be noted that these regressions are necessarily exploratory in nature as they are limited in their statistical

power from only 17–18 observations. Consequently, while we can suggest that our tested explanatory variables are *highly unlikely* to be related to these effects, we cannot definitively conclude that these are wholly unrelated. Even more importantly, our findings should not be interpreted as conclusions related to individual decisionmaking when faced with a financial incentive for vaccination. In fact, our findings suggest that much more research should be done on what considerations, particularly incentives, factor into decisionmaking surrounding vaccination, specifically, and public health behaviors, generally.

## Discussion

Using the synthetic control method, we present empirical evidence supporting: 1) lotteries generally had a positive effect on first-dose vaccinations, 2) lotteries had heavily mixed effects on second-dose vaccinations, and 3) state partisanship, vaccine hesitancy, and the prize-pool had no moderating effect on lotteries and vaccination rates. While other scholars have examined lotteries, none have taken the step we have to look at *every* state which instituted a lottery in support of increasing COVID-19 vaccination rates. Our findings provide some support for lotteries as an incentive tool, in contrast to previous research, but also note important limitations.

First, the lotteries as designed created a significant incentive for vaccine-reluctant individuals to get a first-dose of a vaccine. An individual could only get access to the potential benefits of the lottery by receiving that first-dose; for many individuals, that potential benefit was enough to outweigh the costs which had previously kept them from pursuing vaccination. Yet, once someone gained access to the potential benefit, there was no further incentive to encourage completion of the vaccination regimen. The lotteries had an one entry per person structure, meaning additional shots did not equate to improved odds, marginal though they may be, at securing the jackpot. These findings highlight an important flaw in the incentive design. In order to reach the goal of full vaccination, the incentive structure needs to encourage that behavior; in this case, that would have required additional entries, conditioning entry to the lottery only on full vaccination, or providing a tiered reward structure to encourage vaccination at each step.

Second, and surprisingly, state partisanship was not a clear moderating variable influencing the relationship between vaccination and lottery incentives. Our results demonstrate partisans in the mass public responded to lottery incentives regardless of their own partisanship; no discernible difference, positive or negative, was observed among Republicans or Democrats. While this finding was surprising, it is also potentially encouraging. Even in the midst of unprecedented polarization around a public health threat, the public responded to economic incentives without regard for partisan affiliation. Further research should be conducted to assess how and whether economic incentives could be used to break the polarization on other issues of importance, such as changing behaviors to mitigate climate change, advance anti-racist behaviors, or promoting civic participation.

Third, vaccine hesitancy, at least measured through surveys, appears to have no moderating effect on the effectiveness of these lotteries. Furthermore, the prize-pool also appears to be unrelated. These findings are surprising given what we know about financial incentives, but does suggest avenues for future research. In particular, future work should assess, at an individual level, what odds and prize-pools are most effective at inducement and whether there are significant diminishing marginal effects from increasing the prize-pool. Furthermore, work should also assess whether individual cash-transfers or lotteries are more effective at encouraging vaccination. Finally, if stated vaccine hesitancy is potentially unrelated at an individual

level, then what are the factors are determining whether an individual is induced by a financial incentive, particularly lotteries?

Certainly, lotteries are not a universal cure for improving society's behavior. In some situations, the incentive may not prove sufficient to overcome the costs associated with behavioral change; indeed, our analysis demonstrates an absence of universal behavioral change in a positive direction in each state that instituted a lottery to improve vaccination rates. Nevertheless, economic incentives do have an important place in the toolbox of policy options to address situations where individual behavior changes can lead to positive group benefits. In COVID-19 vaccination, lotteries appear to have helped more than they hurt.

## Supporting information

**S1 Appendix. Synthetic control method details.**
(PDF)

**S2 Appendix. All states' trend-line and control-weights plots.**
(PDF)

**S1 Table. First-dose vaccination MSPE table.**
(PNG)

**S2 Table. Complete vaccination MSPE table.**
(PNG)

**S3 Table. Descriptive statistics of all data.**
(PNG)

## Author Contributions

**Conceptualization:** Sam Fuller, Sara Kazemian, Carlos Algara, Daniel J. Simmons.

**Data curation:** Sam Fuller, Carlos Algara.

**Formal analysis:** Sam Fuller.

**Investigation:** Sam Fuller, Carlos Algara.

**Methodology:** Sam Fuller.

**Project administration:** Sara Kazemian, Carlos Algara, Daniel J. Simmons.

**Software:** Sam Fuller.

**Supervision:** Daniel J. Simmons.

**Validation:** Sam Fuller.

**Visualization:** Sam Fuller, Carlos Algara.

**Writing – original draft:** Sam Fuller, Sara Kazemian, Carlos Algara, Daniel J. Simmons.

**Writing – review & editing:** Sam Fuller, Daniel J. Simmons.

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
