## [Decision Letter · Decision Letter 0]

29 Aug 2022

Assessing the effectiveness of COVID-19 vaccine lotteries: A cross-state synthetic control methods approach

PONE-D-22-08024

Dear Dr. Fuller,

We’re pleased to inform you that your manuscript has been judged scientifically suitable for publication and will be formally accepted for publication once it meets all outstanding technical requirements.

Kind regards,

Tiago Pereira

Academic Editor

PLOS ONE

Additional Editor Comments:

I agree with the other two referees that the paper is well written and the basic statistical analysis seem to support to claims. I therefore, recommend the manuscript for publication.

Reviewers' comments:

Reviewer's Responses to Questions

**Comments to the Author**

1. Is the manuscript technically sound, and do the data support the conclusions?

Reviewer #1: Yes

Reviewer #2: Yes

2. Has the statistical analysis been performed appropriately and rigorously? 

Reviewer #1: Yes

Reviewer #2: Yes

3. Have the authors made all data underlying the findings in their manuscript fully available?

Reviewer #1: Yes

Reviewer #2: Yes

4. Is the manuscript presented in an intelligible fashion and written in standard English?

Reviewer #1: Yes

Reviewer #2: Yes

5. Review Comments to the Author

Reviewer #1: This manuscript addresses the central question of whether financial-based incentives increase the attendance to vaccination programs against covid-19. To answer it, the authors analyze data retrieved from all the states that implemented lotteries to improve vaccination rates. In order to assess the effectiveness of those interventions, a synthetic control method is employed using as covariates demographic data from the U.S. Census data and vaccine hesitancy rates, which were calculated from the raw data from the U.S. Census' Week 29 Household Pulse Survey. The findings put forward in the manuscript can be summarized as follows: i) lotteries did have a positive effect; ii) no significant effect was observed on the number of second doses (as the authors explain, the reason for this resides in the very design of the lotteries); and, perhaps most surprising, iii) state partisanship was shown to have no noticeable influence on the relationship between vaccination and lottery incentives.

As acknowledged in the manuscript, this is not the first study that analyzes the impact of lotteries on vaccination rates. However, the novelty here is that the evaluation is performed taking into account every state that implemented incentive programs. This approach not only provided robust evidence that lotteries can be beneficial for vaccination campaigns, but also highlighted two important facts that might have passed unnoticed in previous works: namely, the possibility that the number of administered second doses could have been improved by modifications in the incentive programs, and the fact that state partisanship did not influence the participation in the lotteries. The former could appear somewhat obvious, but the latter comes as a surprise given the inflamed polarization that has taken over the public health debate. Overall, the work is generally clearly written, concise and easy to read; the results are solid and interesting, and I do not have any major concerns. I therefore recommend the publication.

Reviewer #2: As a mathematician, I do not feel comfortable reviewing this paper in terms of its implications to political policies. Even so, I was able to follow it and understand the statistical analysis. The statistical analysis performed was very basic, but supports the main argument. I recommended it to be accepted based on the statistical analysis, hoping other reviewers were able to assert the quality of its conclusions to political policies.

6. PLOS authors have the option to publish the peer review history of their article (what does this mean?). If published, this will include your full peer review and any attached files.

Reviewer #1: No

Reviewer #2: No

---

## [Editor Report · Acceptance letter]

5 Sep 2022

PONE-D-22-08024 

Assessing the effectiveness of COVID-19 vaccine lotteries: A cross-state synthetic control methods approach 

Dear Dr. Fuller:

I'm pleased to inform you that your manuscript has been deemed suitable for publication in PLOS ONE. Congratulations! Your manuscript is now with our production department. 

Kind regards, 

on behalf of

Dr. Tiago Pereira 

Academic Editor

PLOS ONE